# Evaluating the Impact of Active Footwear Systems on Vascular Health and Static Balance: An Exploratory Study

**DOI:** 10.3390/s25061724

**Published:** 2025-03-11

**Authors:** Susana Lopes, Mário Rodrigues, Mário Lopes, Rui Costa, Joaquim Alvarelhão

**Affiliations:** School of Health Sciences and Institute of Biomedicine, University of Aveiro, 3810-193 Aveiro, Portugal; pslopes@ua.pt (S.L.); mmpr@ua.pt (M.R.); mariolopes77@ua.pt (M.L.); rcosta@ua.pt (R.C.)

**Keywords:** vascular health prevention, footwear, reflection photoplethysmography

## Abstract

Work-related musculoskeletal disorders are prevalent in occupations requiring prolonged standing and repetitive movements, often leading to vascular issues and reduced static balance. Innovations in wearable technology, such as smart footwear integrating active systems, aim to mitigate these challenges. This exploratory study assessed the effects of a novel active footwear system, incorporating compression and vibration, on vascular blood flow and static balance in healthy adults. Sixteen healthy participants (seven men and nine women) were randomized into active and placebo phases, each involving repetitive tasks. Outcomes included reflection photoplethysmography, postural sway, and foot volumetry. Data were analyzed pre- and post-intervention, with statistical significance set at *p* < 0.05. For men, significant improvements in reflection photoplethysmography median values were observed post-active phase (*p* = 0.031), while women showed no change. Enhanced static balance, reflected in decreased total sway (*p* = 0.025), was noted in women. No significant changes occurred during the placebo phase. The active system improved vascular function in men and static balance in women, highlighting its potential for ergonomic interventions in industrial settings. Future studies should explore long-term effects and applications in diverse populations, including those with work-related musculoskeletal disorders.

## 1. Introduction

Work-related musculoskeletal disorders (WRMSDs) are a prevalent concern in industrial environments, where long work hours, sustained low-level tasks, and precise handwork pose significant risks for neck and shoulder complaints, as well as lower limb fatigue and pain [1]. Prolonged standing, a common occupational hazard in sectors such as sales, food service, healthcare, education, and manufacturing, further exacerbates these issues. According to the European Survey of Working Conditions, 47% of workers spend more than 75% of their working hours standing [2].

Beyond posture-related discomfort, excessive lumbar flexion/extension and lower limb asymmetries can disrupt normal body kinetics, leading to altered gait, movement patterns, and balance [1]. The complexity of WRMSDs arises from their multifactorial nature, with influences stemming from physical, psychosocial, and ergonomic factors. Risk factors such as age, gender, job type, and psychosocial stressors have been identified. For instance, female workers and those with permanent contracts exhibit higher odds of developing WRMSDs [3]. Significant associations between WRMSDs and factors such as prolonged standing and repetitive movements further highlight the relevance of these issues [4].

To address these concerns, smart wearable devices are emerging as a promising ergonomic intervention. These devices are designed to deliver targeted stimuli to specific body regions, providing comfort and promoting mobility through lightweight and flexible materials. Technologies such as body mapping and advanced textiles with properties like breathability and flexibility are being incorporated into wearables to enhance user comfort and health [5]. Active wearable devices have shown potential for stimulating muscle activity and preventing vascular problems, especially in areas prone to strain, such as the plantar plexus and ankle.

The growing field of flexible electronics for healthcare applications has encouraged the development of multifunctional wearable fabrics for personal therapy and health management. Such wearable technologies are also increasingly used in occupational settings to prevent musculoskeletal disorders and support workers during physically demanding tasks. This integration of technology into products represents a growing market, driving innovation in healthcare and ergonomic solutions. Industry 4.0 further highlights the importance of incorporating workers as essential elements of the productive environment, utilizing digital solutions to minimize workplace stress and prevent absenteeism.

Maintaining vascular health is crucial to reducing lower limb fatigue. A common WRMSD-related issue is impaired lymphatic drainage or venous return. Currently, available solutions, such as lower limb sleeves, aim to address this by draining stagnant fluid during work or at home. However, these devices can be heavy, bulky, or induce unnatural movement, limiting their usability. In the therapeutic field, medical devices that apply compressive force to specific regions of the foot, ankle, and calf have been developed to increase vascular blood flow and reduce the risk of venous blood clots [6]. There are also commercial systems that use electrical stimuli to target specific muscle groups, inducing isometric contractions to improve circulation [7].

Given the ergonomic demands of industrial work environments, the development of new wearable solutions has gained traction. These devices aim to enhance worker concentration and productivity while addressing comfort and usability, often leveraging textile-based technology for lightweight and adaptable designs. An innovative safety shoe was developed incorporating both sensing and active technologies. This wearable integrates two active mechanisms—compression in the medial arch and supramalleolar vibration—to promote venous drainage, increase balance, and alleviate fatigue. Unlike passive systems, this dynamic approach stimulates the lower limbs to prevent vascular health problems. Therefore, the aim of this exploratory study is to assess the effects of this active system, combining compression and vibration, on static balance and vascular blood flow in healthy participants.

## 2. Materials and Methods

### 2.1. Participants

A convenient sample of 16 healthy adults (9, 56% female) aged 25 to 36 was enrolled. The inclusion criteria for healthy subjects were (i) normotensive, (ii) with a normal body mass index (iii) and no swelling, inflammation, or lesions on the feet or legs, and (iv) the absence of hypertension, peripheral neuropathy, heart disease, or other vascular diseases [8]. Furthermore, all participants were non-smokers, self-referring to regular physical activity, and free of any regular consumption of dietary supplements or medications. Energy drinks (including coffee), alcoholic beverages, and strenuous physical activity were not allowed in the 24 h preceding the experiments [9]. A good sleeping pattern was also assured before the assessment day. All participants provided informed written consent, and all procedures complied with the Helsinki Declaration and relevant regulatory legislation. Data processing adhered to the guidelines set forth by the European General Data Protection Regulation. The study was approved by the ethics committee of the University of Aveiro, Portugal (21-CED/2023).

### 2.2. Study Procedures

Each protocol included a baseline assessment, a 45 min active or placebo phase, and a final assessment moment. The active phase consisted of a repetitive task of the upper limb in a standing position with active footwear. The placebo consisted of using the same footwear but with the active system turned off. A 1 h washout period was assured between the active and placebo phases. The study protocol is presented in Figure 1.

The assignment to the placebo or the active phase was randomized in blocks of two. Both participants and investigators assessing outcomes were blinded to minimize potential bias.

The repetitive task of the upper limb consisted of 5 cycles of 1 min aleatory tapping on 3 blazepods with 1 min rest between cycles. This task was repeated 3 times, with a 5 min period of rest between series, and a total duration of 45 min (Figure 2).

### 2.3. Outcomes

The main outcome was reflection photoplethysmography (PPG), with a Bitalino SENS-PUL-UCE6 sensor (PLUX Biosystems, Lisbon, Portugal—https://support.pluxbiosignals.com/wp-content/uploads/2022/10/Photoplethysmography-PPG-Sensor_Datasheet.pdf (accessed on 3 December 2022)) applied distally on the big toe of the right foot and a sensor at the left ear lobe. Secondary outcomes included static postural sway and foot volumetry. Blood pressure was also assessed prior to each assessment day. All outcomes were assessed before and after both the placebo and active phases. Before the study protocol, demographic and clinical data were collected.

#### 2.3.1. Reflection Photoplethysmography

Measurements were taken in a room with controlled conditions (temperature of 21 ± 1 °C and relative humidity of 40 to 60%) [9]. Participants rested in a lying position for a period of a 10 min before pulse measurements. Three measurements were obtained at each assessment time point and averaged. A meticulous cable screening was assured to reduce contamination by electromagnetic interference. The voltage signal was high-pass filtered to reduce the dominant lower frequency PPG components, leaving the AC pulsatile component for further processing. A low-pass filtering stage with additional electrical mains frequency notch filtering then smoothed the signals. The choice of high-pass filter cut-off frequency is important for PPG measurements as excessive filtering will distort the pulse shape. Typically, the PPG filter bandwidth is chosen to be in the range of 0.05 to 20 Hz. Good measurement techniques should result in reliable and repeatable physiological waveforms, free from movement artifacts or from the significant variability that can be induced by large changes in breathing patterns [10]. The mean values of the last 5 min of each phase of the foot data were used for the analysis of each variable. The variables selected for analysis were the maximum peak, mean peak range, and the root mean square.

#### 2.3.2. Postural Sway

Postural sway was assessed in a 600 × 400 mm force platform (AMTI BP400600-2000, AMTI, Watertown, MA, USA). Participants were instructed to stand as still as possible on the force platform, with their eyes closed, their arms loosely hanging along the body, their knees at full extension, the midpoint of both heels separated by a distance equal to half the foot length, and the forefeet in a free splayed-out position. The feet position was standardized in all trials using a transparent sheet to mark both feet.

Postural sway data were sampled at 1000 Hz (Nexus 1.8.5, Vicon Motion Systems Ltd., Oxford, UK). A custom MATLAB R2014a (MathWorks, Madrid, Spain) program was used for data reduction. The variables derived from the analysis were the total center of pressure (CoP) displacement and CoP velocity.

### 2.4. Active System

The active system consisted of an active device with a compression system located in the medial arch, and a supramalleolar vibration system [11]. The system was integrated into a work boot (ICC, Indústrias e Comércio de Calçado, Guimarães, Portugal)—Figure 3.

The vibration system consisted of piezoelectric motors, with a quadrangular waveform, at 90 Hz. The compression system, the compression mechanism was positioned in the central–medial region, below the metatarsal pad, considering the effective areas to promote the referred stimulus. The system applies a 6 min vibration, followed by a 30 min compression phase and a final 6 min vibration.

### 2.5. Data Analysis

All analyses were conducted with SPSS version 29.0 (SPSS Inc., Chicago, IL, USA). Exploratory data analysis was carried out with the raw values acquired and Shapiro–Wilk tests were performed to determine the normality of the data distribution. The results are presented as mean differences and standard deviations. Comparisons between the beginning and the end of the intervention were determined by Student’s paired *t*-tests. The analyses are presented for the whole sample and by sex. The level of significance was set as *p* < 0.05.

## 3. Results

The difference between before and after the active phase and placebo phase for PPG is presented in Table 1. For the active phase, an increase in the median (*p* = 0.031) of PPG was found for the men. No differences were found for women.

No differences were found for the placebo phase.

Differences were found in postural sway in the active and placebo phases, both before and after the intervention or placebo were carried out. For static balance, a decrease in the total sway (Total CoP displacement) was also found (*p* = 0.025) representing an increase in static balance—Table 2.

## 4. Discussion

In this exploratory study, we examined the effects of an active footwear system—incorporating both compression and vibration mechanisms—on vascular blood flow and static balance in healthy participants. The results demonstrated a tendency for improvement in postural sway for women, indicating increased static balance following the active intervention phase. For vascular function, an increase was observed in the male participants after the active phase for PPG, while no significant differences were found in the placebo phase or for female participants.

These findings align with previous studies that have emphasized the potential of wearable devices in improving postural control and vascular health [5]. Similar devices have shown promise in stimulating muscle activity and improving circulation, particularly in occupational settings where workers are prone to prolonged standing and repetitive motions [7]. Vibrating insoles, for instance, have been shown to improve postural balance and fatigue resistance by stimulating mechanoreceptors in the soles of the feet [12]. By doing so, they help maintain balance and prevent declines in postural stability even in fatigued individuals. This is particularly relevant for workers exposed to prolonged standing conditions, such as those in industrial environments, where postural stability and fatigue play a significant role in musculoskeletal health.

Our results extend the current knowledge by demonstrating the specific effects of a combined compression–vibration system in an industrial work boot. Compression systems located under the metatarsal pad aim to enhance venous return, while vibration systems placed in the supramalleolar (ankle) region improve sensory feedback and proprioception. Together, these mechanisms are designed to address the common ergonomic challenges faced by workers, particularly prolonged standing and reduced circulation.

Interestingly, the improvement in PPG was only significant in the male participants, which may be attributed to gender-specific differences in muscle mass, vascular responses, or the interaction between the active system and individual biomechanics. This observation is consistent with studies that have reported sex-related variations in WRMSD risks and physiological responses to interventions [3]. Men tend to have greater arterial stiffness, and interventions like compression and vibration might more effectively stimulate improved circulation and vascular function in males [13].

The positive effects on static balance can be attributed to the vibration and compression stimuli acting synergistically to stimulate proprioceptive feedback and enhance muscle activation in the lower limbs. The supramalleolar vibration system likely activated mechanoreceptors, which are crucial for maintaining balance and stability [12]. Additionally, the compression mechanism may have improved venous return and reduced lower limb fatigue, allowing for better postural control during the standing tasks.

These effects are consistent with the benefits reported for vibrating insoles in previous studies, where they have been shown to enhance postural stability and reduce the effects of fatigue, particularly in fatigue-prone populations such as military personnel and industrial workers. For example, in previous studies, vibrating insoles were found to prevent significant declines in postural stability, even after fatigue-inducing activities, making them a suitable ergonomic intervention for workers who stand for prolonged periods [12]. One explanation for the better balance improvements seen in women may lie in gender differences in muscle mass and the role of proprioception. Women generally have less muscle mass but are thought to have better fine motor control and proprioceptive feedback, which is essential for maintaining balance. Men, on the other hand, tend to rely more on muscle strength, which might explain why their balance was less affected by the intervention [14].

### 4.1. Implications

These results have important implications for the design of ergonomic interventions in industrial environments. The active footwear system, with its combined compression and vibration mechanisms, presents a promising solution for improving worker health and performance by addressing the common issues related to static postural demands and reduced circulation. Vibrating insoles have demonstrated their potential in work-related environments by mitigating postural instability and reducing fatigue effects [12,15].

This technology could potentially reduce the prevalence of WRMSDs (work-related musculoskeletal disorders), particularly in occupations that require prolonged standing, repetitive movements, and exposure to physically demanding tasks. By combining the benefits of both vibration and compression, this system offers a comprehensive ergonomic solution that addresses both musculoskeletal health and vascular health.

### 4.2. Limitations

Several limitations of this study should be acknowledged. First, the sample size was relatively small, limiting the generalizability of the findings. Additionally, the study only included healthy participants, which may not fully reflect the potential effects of the active system on individuals with existing musculoskeletal disorders or poor vascular health. Moreover, the study protocol was relatively short-term, and future research should consider long-term follow-up to evaluate the sustained effects of the active footwear system.

The lack of significant changes in PPG for the women could be attributed to the short duration of the intervention or the small sample size. It is possible that longer exposure to the active system, or a larger cohort, may yield more pronounced effects on vascular health.

### 4.3. Future Directions

Future studies should investigate the long-term effects of the active system in larger and more diverse populations, including individuals at higher risk of WRMSDs. Additionally, exploring the specific mechanisms underlying the sex differences observed in the PPG results could provide valuable insights for tailoring interventions. Incorporating real-world workplace assessments may also offer practical insights into how this technology can be optimized for industrial settings.

## 5. Conclusions

In conclusion, this study highlights the potential benefits of an active compression and vibration system for improving static balance and enhancing vascular health in healthy individuals. While the results are promising, further research is needed to validate these findings in broader populations and to explore the long-term effects of such ergonomic interventions in occupational environments.

## 6. Patents

This work was conducted under a provisory patent register of the active system described in the methods section.

## Figures and Tables

**Figure 1 sensors-25-01724-f001:**
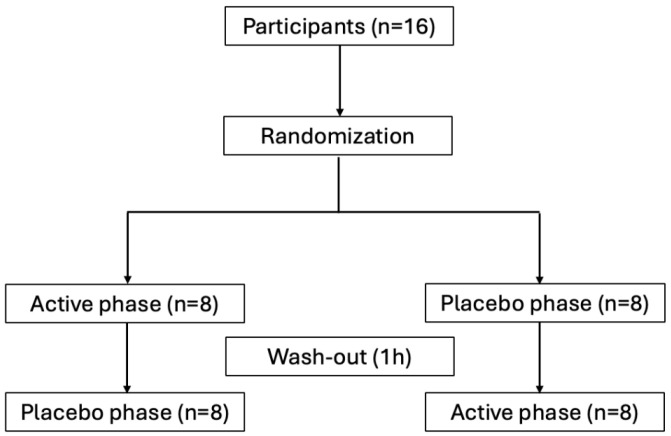
Study protocol design.

**Figure 2 sensors-25-01724-f002:**
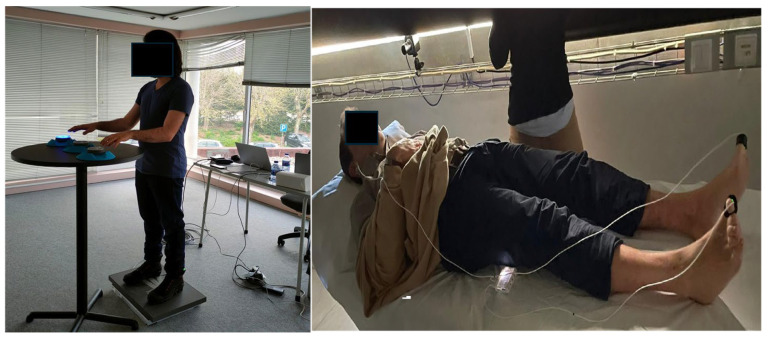
The repetitive task of the upper limb (**left**) and the location of the PPG sensor (**right**).

**Figure 3 sensors-25-01724-f003:**
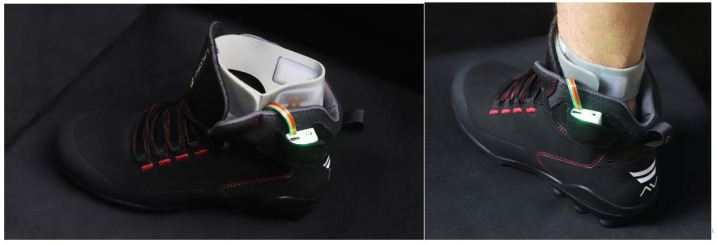
The active system integrated into a work boot.

**Table 1 sensors-25-01724-t001:** The difference between before and after the active phase and placebo phase in reflection photoplethysmography for men and women (raw data).

	Male (n = 7)		Female (n = 9)	
	Difference	*p*-Value	Difference	*p*-Value
Active phase				
Maximum	−44.9 ± 116.1	0.171	14.4 ± 153.2	0.695
Median	4.5 ± 13.8	0.031	−2.4 ± 22.4	0.654
Root Mean Square	−2.1 ± 6.5	0.167	3.3 ± 10.3	0.187
Placebo phase				
Maximum	−7.0 ±183.6	0.889	−60.2 ± 139.5	0.085
Median	−1.9 ± 23.7	0.770	1.9 ± 23.1	0.729
Root Mean Square	1.1 ± 11.7	0.728	−1.8 ± 10.9	0.500

**Table 2 sensors-25-01724-t002:** The difference in postural sway before and after the intervention in the active and placebo phases, for men and women.

	Male (n = 7)		Female (n = 9)	
	Difference	*p*-Value	Difference	*p*-Value
Active phase				
Total CoP ^1^ displacement (mm)	0.016 ± 0.039	0.351	−0.015 ± 0.016	0.025
CoP ^1^ velocity (mm/s)	0.003 ± 0.004	0.903	−0.005 ± 0.010	0.179
Placebo phase				
Total CoP ^1^ displacement (mm)	0.014 ± 0.038	0.312	−0.007 ± 0.018	0.273
CoP ^1^ velocity (mm/s)	0.000 ± 0.004	0.119	0.002 ± 0.007	0.393

^1^ Center of pressure.

## Data Availability

Data will be available for sharing upon reasonable requests with the corresponding author.

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
