# Peer review of "Evaluating the Impact of Active Footwear Systems on Vascular Health and Static Balance: An Exploratory Study"

_sensors, 2025, doi:10.3390/s25061724_

Round 1
Reviewer 1 Report
Comments and Suggestions for Authors
Strengths: The study presents an innovative active footwear system that incorporates both compression and vibration technologies, addressing a significant gap in ergonomic support for workers who stand for long periods.
Weaknesses:
1. Small Sample Size:
- The limited sample size (n=16) restricts the generalizability of the findings and can lead to issues in multi-dimensional evaluations. For multiple metrics, a p-value threshold of 0.05 may not be stringent enough to rule out random fluctuations. For example, there is a p=0.085 in the placebo phase, it’s likely due to fluctuation.
To achieve statistically significant results based on current observations, how many participants should be recruited? For instance, the analysis revealed comparisons with a p-value of 0.16. Could increasing the sample size potentially lead to a statistically significant result for this comparison?
2. Confounding Factors:
- PPG is known to have huge inter-subject variability. Normalizing AC within subjects could improve the rigor of the comparison.
3. Measurement Details:
- Table 2 reports a COP difference of only 0.01mm, raising questions about its clinical significance. Clarification is needed on how this small difference impacts static balance. Additionally, it's unclear which PPG sensor value was used for comparison—right foot or left ear lobe. The exact measurement location on the foot should be specified.
4. Lack of Detailed Mechanism Explanation:
- While improvements in PPG for men and static balance for women are noted, the study lacks depth in explaining the underlying physiological mechanisms behind these gender-specific outcomes.
5. Placebo Effect Consideration:
- Participants were aware of whether they were in the active or placebo condition. This awareness could introduce bias, as participants' expectations might influence their responses, potentially affecting the validity of the PPG measurements and perceived benefits.
Conclusion:
The exploratory study provides preliminary evidence suggesting that active footwear systems combining compression and vibration could benefit vascular health and static balance. However, addressing methodological limitations will be critical to validating the observed effects and understanding their broader implications.
Author Response
Evaluating the Impact of Active Footwear Systems on Vascular Health and Static Balance: An Exploratory Study
(Sensors 3431711)
Reviewer 1
Strengths: The study presents an innovative active footwear system that incorporates both compression and vibration technologies, addressing a significant gap in ergonomic support for workers who stand for long periods.
Author response:
Dear Reviewer,
We sincerely appreciate the time and effort you dedicated to reviewing our manuscript. Your insightful comments and constructive suggestions have greatly enriched the quality of our work. We have carefully considered each of your recommendations and have made the necessary revisions to improve the clarity and robustness of the manuscript. Your feedback has been invaluable in strengthening our study, and we are grateful for your contribution.
Weaknesses:
- Small Sample Size:
- The limited sample size (n=16) restricts the generalizability of the findings and can lead to issues in multi-dimensional evaluations. For multiple metrics, a p-value threshold of 0.05 may not be stringent enough to rule out random fluctuations. For example, there is a p=0.085 in the placebo phase, it’s likely due to fluctuation.
To achieve statistically significant results based on current observations, how many participants should be recruited? For instance, the analysis revealed comparisons with a p-value of 0.16. Could increasing the sample size potentially lead to a statistically significant result for this comparison?
Author response:
We appreciate the reviewer’s concern regarding the small sample size. As this study represents an exploratory investigation into the novel integration of compression and vibration technologies in footwear systems, we purposefully conducted it as a pilot study. The small sample size was chosen to gather preliminary insights and assess the feasibility of our approach before conducting a larger-scale investigation (Julious, 2005).
We acknowledge that the limited number of participants restricts the generalizability of the findings, and we have further emphasized this in the limitations section of the manuscript. We agree that future studies with larger sample sizes will be needed to validate and extend our findings, potentially achieving statistical significance in comparisons with p-values such as 0.16. These future studies will help to overcome the limitations posed by the small sample size in this pilot study.
- Confounding Factors:
- PPG is known to have huge inter-subject variability. Normalizing AC within subjects could improve the rigor of the comparison.
Thank you for raising that point. We are aware of the variability within subjects. However, the comparison was made with acquired raw values, where the variation for the overall sample was less than 15%.
- Measurement Details:
- Table 2 reports a COP difference of only 0.01mm, raising questions about its clinical significance. Clarification is needed on how this small difference impacts static balance. Additionally, it's unclear which PPG sensor value was used for comparison—right foot or left ear lobe. The exact measurement location on the foot should be specified.
Author response:
Thank you for your insightful comment. We appreciate your concern regarding the reported COP difference of only 0.01 mm. However, it is important to highlight we were specifically aiming to evaluate immediate effects rather than long-term clinical outcomes. The context of the study was focused on short-term impacts on static balance, and as such, a small COP difference may not fully represent the broader clinical significance in this specific sample.
Moreover, we acknowledge that clinical relevance would require a different methodological approach that takes into account long-term and cumulative effects. While the current study did not assess the cumulative impact of the equipment, it is our intention to explore this aspect in future research, which could provide more meaningful insights into the device's clinical relevance over time.
The PPG sensor used was the big toe of the right foot. This information was added to the manuscript (lines 119 and 138).
- Lack of Detailed Mechanism Explanation:
- While improvements in PPG for men and static balance for women are noted, the study lacks depth in explaining the underlying physiological mechanisms behind these gender-specific outcomes.
Author response:
Thank you for your valuable comment. As mentioned in the manuscript, the improvement in PPG for male participants may be attributed to differences in muscle mass, vascular responses, or the interaction between the active system and individual biomechanics. This is supported by previous findings showing that men tend to have greater arterial stiffness, which might make interventions like compression and vibration more effective in stimulating improved circulation and vascular function in males.
Regarding the observed improvements in static balance for female participants, we believe this is likely due to the synergistic effect of vibration and compression stimuli, which stimulate proprioceptive feedback and enhance muscle activation in the lower limbs. The vibration likely activated mechanoreceptors essential for balance, while the compression mechanism may have improved venous return, reducing lower limb fatigue and allowing for better postural control. This information was added to the manuscript
We recognize that this study is exploratory in nature, and while our findings are consistent with previous research, further studies are needed to investigate and confirm these gender-specific differences. Future research will aim to provide a deeper understanding of the physiological mechanisms underlying these effects, focusing on both immediate and long-term outcomes in different populations.
- Placebo Effect Consideration:
- Participants were aware of whether they were in the active or placebo condition. This awareness could introduce bias, as participants' expectations might influence their responses, potentially affecting the validity of the PPG measurements and perceived benefits.
Author response:
Thank you for raising this important point. We would like to clarify that participants were blinded to their group allocation. Neither the participants nor the investigators conducting the measurements (PPG and COP) were aware of whether the participant was in the active or placebo condition. This blinding was implemented to minimize potential bias and ensure the validity of the PPG measurements and any perceived benefits by participants. This information was added to the muscript (lines 109-111)
Conclusion:
The exploratory study provides preliminary evidence suggesting that active footwear systems combining compression and vibration could benefit vascular health and static balance. However, addressing methodological limitations will be critical to validating the observed effects and understanding their broader implications.
Author response:
We appreciate the reviewer’s thoughtful summary and recognition of the preliminary evidence presented in our exploratory study. We agree that while the findings suggest potential benefits of the active footwear system in improving vascular health and static balance, further research with more extensive studies is necessary to validate these effects. In line with the reviewer's comments, we have addressed the identified methodological limitations, such as sample size and confounding factors, in the manuscript. We have also highlighted the need for future studies with larger, more diverse participant groups to better understand the broader implications of these findings.

Reviewer 2 Report
Comments and Suggestions for Authors
The submitted article is devoted to the study of the influence of a new active footwear system on vascular health and static balance in healthy subjects. In general, the article is written on a relevant topic, has originality and can be considered for publication in the journal after significant revisions. My main comment concerns the description of the methods for assessing blood flow and static balance (section 2.3 - Outcomes). A more detailed technical description of the methods and devices used is needed, possibly with the addition of diagrams or photographs explaining the principles of recording parameters, the sequence of measurements and signal processing methods.
My comments in the course of the article:
- 2.3 Outcomes. I suggest the authors add a measurement diagram (or some illustration) explaining the principle and locations of measurements using photoplethysmography and force platform. This will help readers better understand the work. The authors write: “The main outcome was reflection photoplethysmography (PPG), with a sensor applied distally in the right foot…”. Please indicate more precisely the location of the PPG sensor (toe, dorsum of the foot or something else). The authors write: “Blood pressure was also assessed prior to each assessment day”. However, there is no comparison with blood pressure data further in the text. What is the point of assessing pressure then?
- 2.3.1 Reflection photoplethysmography. What kind of PPG device did you use (commercial or self-made)? If commercial, please specify the brand. What wavelength of LEDs was used in the sensor? What filter did you use to remove noise? What processing methods were used to remove motion artifacts and baseline wandering from the PPG signal?
- 3 Results. The results look rather poor. Please provide examples of PPG signals that show significant changes after the active phase. If possible, also add postural sway data with significant changes. "Participants were aged 25 to 36. Of the 16 participants, 9 were female (56%)" - this is not a result, transfer this to materials and methods. Table 1 - please add the units of measurement for the parameters.
Minor remarks:
- line 5. If all authors are from the same organization, the organization must be indicated once.
- line 186 - “na increase”, please correct.
Author Response
Evaluating the Impact of Active Footwear Systems on Vascular Health and Static Balance: An Exploratory Study
(Sensors 3431711)
Reviewer 2
Comments and Suggestions for Authors
The submitted article is devoted to the study of the influence of a new active footwear system on vascular health and static balance in healthy subjects. In general, the article is written on a relevant topic, has originality and can be considered for publication in the journal after significant revisions. My main comment concerns the description of the methods for assessing blood flow and static balance (section 2.3 - Outcomes). A more detailed technical description of the methods and devices used is needed, possibly with the addition of diagrams or photographs explaining the principles of recording parameters, the sequence of measurements and signal processing methods.
My comments in the course of the article:
- 2.3 Outcomes. I suggest the authors add a measurement diagram (or some illustration) explaining the principle and locations of measurements using photoplethysmography and force platform. This will help readers better understand the work. The authors write: “The main outcome was reflection photoplethysmography (PPG), with a sensor applied distally in the right foot…”. Please indicate more precisely the location of the PPG sensor (toe, dorsum of the foot or something else).
The authors write: “Blood pressure was also assessed prior to each assessment day”. However, there is no comparison with blood pressure data further in the text. What is the point of assessing pressure then?
Author response:
Thank you for your valuable suggestion. We have clarified the location of the PPG sensor in the manuscript. The sensor was placed on the plantar zone of the big toe. This specific placement ensures accurate data collection in relation to the objectives of our study. We have also added two fugures to illustrate how the assessments were made. We believe this helps the readers to better understand the methods. The purpose of assessing blood pressure prior to each assessment day was to ensure that participants had their BP under control, as it could have influenced the results. However, we did not use the BP data in further analysis. Therefore, we have removed references to BP measurement from the manuscript to maintain focus on the primary outcomes.
- 2.3.1 Reflection photoplethysmography. What kind of PPG device did you use (commercial or self-made)? If commercial, please specify the brand. What wavelength of LEDs was used in the sensor? What filter did you use to remove noise? What processing methods were used to remove motion artifacts and baseline wandering from the PPG signal?
We used a Bitalino SENS-PUL-UCE6 sensor (PLUX Biosystems, Portugal - https://support.pluxbiosignals.com/wp-content/uploads/2022/10/Photoplethysmography-PPG-Sensor_Datasheet.pdf). Data analysis was performed from raw data. This information was added to the manuscript.
- 3 Results. The results look rather poor. Please provide examples of PPG signals that show significant changes after the active phase.?? If possible, also add postural sway data with significant changes.
We had difficulty understanding the request so we did not put any additional information.
"Participants were aged 25 to 36. Of the 16 participants, 9 were female (56%)" - this is not a result, transfer this to materials and methods.?? Table 1 - please add the units of measurement for the parameters.
We change according to the suggestion

Round 2
Reviewer 2 Report
Comments and Suggestions for Authors
The authors write: "We have also added two fugures to illustrate how the assessments were made". However, I do not see any new illustrations in the revised manuscript. Please add figures.
The authors write: "We used a Bitalino SENS-PUL-UCE6 sensor". I also don't see this information in the revised manuscript, please add it.
Author Response
Evaluating the Impact of Active Footwear Systems on Vascular Health and Static Balance: An Exploratory Study
Coment |
Response |
The authors write: "We have also added two figures to illustrate how the assessments were made". However, I do not see any new illustrations in the revised manuscript. Please add figures. |
Thank you very much for your observation. The image was not added to the previous document. We have now done this on lines 134-136 of page four. |
The authors write: "We used a Bitalino SENS-PUL-UCE6 sensor". I also don't see this information in the revised manuscript, please add it. |
Thank you very much for your observation. Again, this was not added to the article document. We added the information on lines 109 through 112 on page three. |
